# A Biological Feature and Heterogeneous Network Representation Learning-Based Framework for Drug–Target Interaction Prediction

**DOI:** 10.3390/molecules28186546

**Published:** 2023-09-09

**Authors:** Liwei Liu, Qi Zhang, Yuxiao Wei, Qi Zhao, Bo Liao

**Affiliations:** 1College of Science, Dalian Jiaotong University, Dalian 116028, China; liutree80@163.com (L.L.); qiz991003@163.com (Q.Z.); 2Key Laboratory of Computational Science and Application of Hainan Province, Hainan Normal University, Haikou 571158, China; 3College of Software, Dalian Jiaotong University, Dalian 116028, China; w635312612@163.com; 4School of Computer Science and Software Engineering, University of Science and Technology Liaoning, Anshan 114051, China

**Keywords:** drug–target interactions, graph convolutional network, graph attention network, representation learning, machine learning

## Abstract

The prediction of drug–target interaction (DTI) is crucial to drug discovery. Although the interactions between the drug and target can be accurately verified by traditional biochemical experiments, the determination of DTI through biochemical experiments is a time-consuming, laborious, and expensive process. Therefore, we propose a learning-based framework named BG-DTI for drug–target interaction prediction. Our model combines two main approaches based on biological features and heterogeneous networks to identify interactions between drugs and targets. First, we extract original features from the sequence to encode each drug and target. Later, we further consider the relationships among various biological entities by constructing drug–drug similarity networks and target–target similarity networks. Furthermore, a graph convolutional network and a graph attention network in the graph representation learning module help us learn the features representation of drugs and targets. After obtaining the features from graph representation learning modules, these features are combined into fusion descriptors for drug–target pairs. Finally, we send the fusion descriptors and labels to a random forest classifier for predicting DTI. The evaluation results show that BG-DTI achieves an average AUC of 0.938 and an average AUPR of 0.930, which is better than those of five existing state-of-the-art methods. We believe that BG-DTI can facilitate the development of drug discovery or drug repurposing.

## 1. Introduction

The determination of drug–target interaction (DTI) is of great significance for the development of new drugs and the understanding of drug side effects. There are currently tens of thousands of Food and Drug Administration-approved drugs on the pharmaceutical market, as well as new drugs that are being validated in clinical trials [1]. These new drugs may interact with potential unseen targets, treat unknown diseases, and produce certain side effects. Although traditional biochemical experiments can accurately verify DTI, the identification of DTI using biochemical experiments is a time-consuming, laborious, and expensive process [2]. In order to accelerate the development of new drugs and reduce the workload of laboratory experiments, it is important to establish an effective DTI recognition model. The existing computation-based methods for predicting DTI mainly include the following two types: biological feature-based methods and network-based methods.

The first type is biological feature-based methods. The main idea of these methods is to extract the features of drugs and targets through their biological sequences. Based on the extracted features, a deep learning model is used for DTI recognition. Bleakley et al. converted the DTI recognition problem into a binary classification problem by using the binary local model method [3]. In 2017, Meng et al. proposed a model called PDTPS [4], which extracted features from protein sequences and medicinal chemical structures and applied relevance vector machines [5] to predict DTI. In 2018, Wang et al. developed a stacked autoencoder-based model [6], which extracted features of proteins by way of a position-specific scoring matrix [7] and applied a random forest algorithm to predict DTI. In 2022, Cheng et al. proposed a model for predicting DTI using interaction and independent features based on the attentional mechanism [8]. Specifically, protein sequences were extracted by using multiscale one-dimensional convolution, and word2vec was used for pre-training during word embedding. The RDKit tool was used to convert the SMILES sequence into a graph structure and input it into the multi-layer graph attention network (GAT) for drug feature extraction. In the same year, Zhao et al. constructed a model named HyperAttentionDTI [9], which applied biological sequences to deep learning models with attentional mechanisms to predict DTI. A one-dimensional convolution of three stacked layers was used to learn sequence features from the inputs. The attention mechanism module produced an attention fraction value for each pair of amino acids. After the attention module, the multilayer fully connected neural network (FCNN) was used for the prediction of the DTI. In 2023, Bai et al. introduced a model named DrugBAN [10], which extracted features from drug molecular diagram and target protein sequences through a graph convolutional network (GCN) and a 1D convolutional neural network (CNN), and used FCNN to predict DTI. Some biological features can be extracted from literature information; such approaches use drug and target descriptions as features rather than biological sequences. In 2015, Alkema et al. introduced important techniques for text mining such as askMEDLINE, PubNet, PubViz, CoPub, etc. [11]. These methods can be used to effectively analyze the growing number of research papers in the field of bioinformatics. In 2016, Fu et al. presented a machine learning model based on semantic similarity to identify DTIs [12].

The second type is the network-based approach. Networks can describe complex and diverse relationships between drugs and proteins. The main idea of these methods is to extract features by constructing an interaction network and a similarity network. In 2017, Luo et al. proposed a model named DTINet [13], which established heterogeneous networks by obtaining multiple drug-related information and protein-related information, and predicted DTI through the networks. Not long after this, Yan et al. developed a model that extracted features from heterogeneous information on drugs and targets and applied multi-kernel learning and clustering methods to predict DTI [14]. In 2020, Zhao et al. introduced a model that extracted features from a drug–protein pair network and combined GCN and a deep neural network to predict DTI [15]. In 2021, Peng et al. presented a model named EEG-DTI [16]. Specifically, they constructed a complex heterogeneous network containing drugs, proteins, diseases, and side effects. The features of drugs and targets were performed using a three-layer GCN learning framework. Finally, the inner product method was used to predict DTI. In the same year, An et al. developed a model named NEDTP [17], which is a similarity-based method for predicting DTI. The author constructed a similar network of nodes through 15 heterogeneous information networks including drug–drug interaction, drug–disease association, drug–side effect association, the gene ontology biological process, protein–protein interaction, protein–disease association, protein sequence similarity, etc. The second-order biased random walk algorithm and word2vec were used for feature sampling and node vector representation learning. Finally, the model used a gradient boosting decision tree [18] to predict DTI. In 2022, Li et al. combined a transformer module with a communicative message passing neural network to better capture the two-way effects between drugs and targets, and predicted DTI using multilayer perceptron [19].

The relationship between drugs and targets is complex and varied. We should consider the features of drugs and targets in multiple networks, such as drug–disease associations, drug–drug similarities, drug–drug interactions, drug–side effect associations, protein–protein similarities, and disease–protein associations. The existing methods usually do not consider interaction types sufficiently and therefore do not take into account the relationship between multiple biological entities well. In some methods, drugs, diseases, proteins, side effects, etc. are taken as network nodes to obtain more correlational relationships when constructing heterogeneous networks.

However, there are some obvious disadvantages to the previous methods. First, these methods do not make full use of the biometric features of drugs, targets, and abundant topological information in heterogeneous networks to predict DTI. Specifically, biological feature-based methods do not take into account the features of drugs and targets in heterogeneous networks containing multiple biological entities (i.e., drugs, proteins, diseases, side effects), nor how to extract and incorporate them. Network-based methods only consider the topological information of drug target pairs and do not comprehensively consider the biological structure information of drugs and targets. Increasing node types also creates the problem of increasing computational complexity and makes the model more dependent on the existing data, which may eventually reduce the generalization ability of the model. Second, the predictive classification process of these methods is often fused with the feature extraction module to form an end-to-end DTI prediction model [20,21,22], but these methods also reduce the flexibility and interpretability of the model and sometimes introduce overfitting problems.

To solve the existing difficulties, we propose a new framework for predicting DTI based on biological feature and graph representation learning, named BG-DTI, to identify the interactions between drugs and targets. In our study, we first extract biological sequence information on drugs and targets using sequence embedding, a CNN layer and a pooling layer, and then consider both biological sequence information and network information and add drug–drug similarity and target–target similarity to complete the heterogeneous network. Based on heterogeneous networks, we use a combination method of GCN and GAT to learn the features representation of drugs and targets. Lastly, we use the random forest classifier to predict whether there is an interaction between drugs and targets. After model construction, we conduct a five-fold cross-validation (5-fold CV) experiment to evaluate the performance of BG-DTI. Meanwhile, we also compare BG-DTI with state-of-the-art methods on a benchmark dataset. All the results show that BG-DTI performs better than existing state-of-the-art methods, and it is an efficient model for DTI prediction.

## 2. Results

### 2.1. Performance Evaluation

The drug–target heterogeneous network data often have the problems of large positive and negative sample disparity and unbalanced samples, which will affect the effect and stability of the model. We conduct over-sampling and under-sampling of positive and negative samples, respectively, to balance samples after sampling, thus improving the generalization ability of BG-DTI. We compare the effect of the model on various sampled datasets while keeping other parameters values unchanged. The final parameters of BG-DTI are shown in Appendix A Table A1. In model evaluation, the area under the receiver operating characteristics curve (AUC) and the precision–recall curve (AUPR) are used as the model evaluation criteria [23,24]. Among them, the receiver operating characteristics (ROC) curve abscissa and ordinate are FPR (false positive rate) and TPR (true positive rate), respectively. The precision–recall (PR) curve abscissa and ordinate are precision rate and recall rate, respectively. The formulas of these metrics are shown as follows:(1)$$TPR=\frac{TP}{TP+FN}$$
(2)$$FPR=\frac{FP}{TN+FP}$$
(3)$$Precision=\frac{TP}{TP+FP}$$
(4)$$Recall=\frac{TP}{TP+FN},$$
where TP (True Positive) and TN (True Negative) are the number of correctly predict positive and negative samples, respectively. FP (False Positive) and FN (False Negative) are the number of wrongly predicted positive and negative samples, respectively.

The results are shown in Figure 1. When over-sampling and under-sampling are used, the model shows a significant improvement in the AUC and AUPR indexes. Compared with the model without positive sample over-sampling, the AUC of BG-DTI with positive sample over-sampling increases by 5.1%. In AUPR performance, it increases by 5.0%. This indicates that the data balance in the network can improve the generalizability of the model and improve the stability of the model. We use the 5-fold CV method to evaluate the model. In order to ensure the accuracy of the experimental results, the following test results are the average value of 30 runs. We show the performance of 5-fold CV in Table 1. When the number of folds is five, the average AUC and AUPR of BG-DTI in the benchmark dataset reach 0.938 and 0.930, respectively.

### 2.2. Comparison with Previous Methods

To demonstrate the superiority of our model, we use 5-fold CV to compare the performance of BG-DTI and five other methods, namely MultiDTI [25], HyperAttentionDTI [9], NeoDTI [26], DTINet [13], and HNM [27] on the benchmark dataset. Table 2 shows the performance comparison of BG-DTI and comparative models, and the specific descriptions of these comparative models are as follows:MultiDTI is a machine learning method based on multi-modal representation learning. It integrates heterogeneous network information between new chemical entities to predict DTI.HyperAttentionDTI is a sequence-based deep learning model. It uses attention mechanisms to improve the prediction of DTI and narrow the search space for drug and target binding sites.NeoDTI is a method using a graph neural network, which can integrate multiple information sources and automatically learn the topological structure information of each node vector in the network.DTINet is a deep learning method for predicting DTI. It uses random walk and singular value decomposition to compute the embedding vectors of the drug and target. DTINet identifies the DTIs based on these embedding vectors.HNM is a computational framework based on a heterogeneous network model. The model calculates the intensity between disease–drug pairs using an iterative algorithm on the heterogeneous graph, which also contains drug target information.

For a fair comparison, we adopt the default parameter values of the respective original implementations for the five methods and compare them on the benchmark dataset. It can be intuitively shown that BG-DTI performs better than the existing state-of-the-art methods from Table 2, in which both AUC and AUPR of our model are above those of other models. Specifically, BG-DTI achieves an average AUC of 0.938, which is 14.7%, 1.1%, 2.4%, 4.8%, and 4.9% higher than that of MultiDTI, NeoDTI, DTINet, HNM, and HyperAttentionDTI, respectively. The average AUPR of BG-DTI is 0.930, which is 7.5%, 5.6%, 0.7%, 35.8%, and 3.3% higher than that of MultiDTI, NeoDTI, DTINet, HNM, and HyperAttentionDTI. From the results, we can see that the predictive accuracy of HNM is much lower than that of BG-DTI, and the reason may be that HNM does not obtain enough topological structure information and heterogeneity information in heterogeneous networks. DTINet outperforms other models and obtains the second-best performance, which may be because this model further introduces medicinal chemical structure information and protein sequence information into the heterogeneous network by creating two additional similarity networks. In summary, the evaluation results show that BG-DTI performs better than existing state-of-the-art methods.

### 2.3. Performance on the Cold Start Validations

The task of DTI is to find new drugs that interact with existing targets in the real world. In practice, the number of known targets is limited, and most of them are used for model training, which means that the cold start validations are closer to the situation in the real world. To reflect the challenging situations in which the predictor will be used in practice, we conduct three cold start validations. Table 3 shows the performance comparison of BG-DTI with other methods in terms of three settings from the same benchmark dataset with 5-fold CV, and the three settings are as follows:Blind drug: There is no overlap of drugs between the training and test datasets.Blind protein: There is no overlap of proteins between the training and test datasets.Blind pair: There is no overlap between the training and test datasets. None of the drugs and proteins in the training datasets are present in the test datasets.

As shown in Table 3, BG-DTI shows remarkable performances in terms of the three cold start validations. In the blind drug test, the AUC score of BG-DTI is 0.911 and the AUPR score is 0.887, whereas the competitors perform poorly. In the blind protein test, we discover no significant difference between the five comparative models and BG-DTI. Nevertheless, BG-DTI performs slightly better than the other models in terms of AUC score and is barely behind the top-ranked model in terms of AUPR score. With reference to the blind pair setting, the AUC score of BG-DTI is 0.904, which is 8.4% higher than the best baseline HyperAttentionDTI (0.820) and 10.0% higher than that of DTINet (0.804). The AUPR score of BG-DTI is 0.875, which is 5.7% higher than the best baseline DTINet (0.818) and 7.4% higher than that of MultiDTI (0.801). In general, our model achieves a reasonable performance on three cold start validations, indicating that BG-DTI trained on the benchmark dataset generalizes well to independent virtual screening tasks and deals with the challenges in real experiments.

### 2.4. Ablation Experiments

To evaluate the effects of biological features, heterogeneous networks, node similarity, and the prediction module in BG-DTI, we conduct ablation studies using 5-fold CV on the benchmark dataset, to evaluate the impact of four components on the predictive performance of BG-DTI. In the experiment, we keep the default parameter values unchanged. DTI-bio: we remove the biometric feature extraction module, and the input features of the nodes in heterogeneous networks are modified to the random generation vector.DTI-het: we remove the graph representation learning model of drug and target in BG-DTI.DTI-sim: we remove the drug–drug similarity information and protein–protein similarity information in the heterogeneous network.DTI-rf: we apply multilayer perceptron instead of the random forest prediction module, constituting an end-to-end DTI prediction model. The classification score threshold is set to 0.5.

Table 4 shows the performance comparison of BG-DTI and its four variants in terms of AUC and AUPR on the same benchmark dataset. The results show that the performance of BG-DTI is better than that of DTI-bio, DTI-het, DTI-sim, and DTI-rf. BG-DTI obtains the best AUC score of 0.938, and it is 5.7%, 6.3%, 6.1%, and 9.7% higher than that of DTI-bio, DTI-het, DTI-sim, and DTI-rf, respectively. The AUPR score of BG-DTI is 0.930, which is 4.4%, 4.1%, 5.0%, and 20.7% higher than that of DTI-bio, DTI-het, DTI-sim, and DTI-rf, respectively. It indicates that these parts in our design can improve the predictive performance. The performance of both DTI-bio and DTI-het are lower than that of BG-DTI, suggesting that biometric features extraction and graph representation learning of the drug–target are necessary for predicting DTI. Furthermore, it is worth noting that the AUPR of DTI-rf is 20.7%, 16.3%, 16.6%, and 15.7% lower than that of BG-DTI, DTI-bio, DTI-het, and DTI-sim, respectively. The reason is that DTI-rf correctly predicts more negative samples and a large proportion of positive samples are also predicted as negative. This shows that the random forest algorithm can effectively improve the recognition performance of DTI, and the end-to-end model may make the prediction more dependent on the composition and distribution of training data.

## 3. Materials and Methods

### 3.1. Datasets

Currently, the most widely used DTI datasets are the Luo et al. dataset [13], the Yamanishi et al. dataset [28], and the Davis dataset [29]. By deleting the overlapping data with little correlation in the above three datasets, we make the remaining data constitute a complete dataset. The new dataset contains six related networks (drug–disease association network, drug–drug interaction network, drug–protein interaction network, drug–side effect association network, protein–disease association network, protein–protein interaction network), and drug and protein sequences. Of these, 598 drug nodes represented by SMILES sequences and 1352 original protein sequences represented by amino acid sequences are extracted from the Drugbank database, the PubChem database, and the UniProts database. From the Davis database, 4862 disease nodes are obtained. From SIDER database 2.0, 3640 side effect nodes are extracted [30]. All related network data are given as Boole matrices. That is, 1 indicates a known interaction or correlation and 0 indicates an unknown or no interaction or correlation. The information of our benchmark dataset is shown in Table 5.

### 3.2. Overview of BG-DTI

In this part, we present BG-DTI, a biological feature and heterogeneous network representation learning-based framework for drug–target interaction prediction. The workflow of BG-DTI is illustrated in Figure 2, which includes three modules for model construction: a drug and target biometric feature extraction module, a graphical representation of the learning module, and the DTI prediction module. The detailed descriptions of these modules are introduced as follows. First, we take the SMILE sequence of the drug and amino acid sequence of the target as input, and fully extract adjacent information from the sequence through sequence embedding, CNN layer, and pooling layer, so as to shorten the feature length of the sequence and reduce the computational complexity and time while preserving the original association relationship and global information in the sequence. Second, we construct a drug–target heterogeneous network through the association relationship among various biological entities. In addition to the drug–drug interaction edge, the drug–protein interaction edge, and the protein–protein interaction edge, we also add an additional drug–drug similarity edge and a protein–protein similarity edge to the heterogeneous network. We then stack a GCN layer and a GAT layer to enlarge the receptive field on the graph, effectively aggregate information of multi-hop neighbors, and learn the features of drugs and targets from the heterogeneous network. In order to prevent the loss of feature information, we combine the output of the first and third layers to obtain the final representation of drug and target features. Last, we splice the characteristic representations of drugs and targets to obtain the fusion descriptors of drug–target pairs. Finally, the random forest algorithm is used to predict DTI.

### 3.3. Drug and Target Biological Feature Extraction

Each protein sequence $T=a_{1},a_{2},\cdots ,a_{n},$ where $a_{i}$ represents the *i*th amino acid and *n* is the sequence length. In order to fully extract the amino acid adjacent information in the sequence, we represent every *k* adjacent amino acids as a one-hot encoding. For example, if *k* is 3 and *n* is 5, then the protein sequence *T* is expressed as $\left(a_{1},a_{2},a_{3}\right)$, $\left(a_{2},a_{3},a_{4}\right)$, $\left(a_{3},a_{4},a_{5}\right)$. In this work, each protein original embedding is composed of $20^{3}=800$ one-hot encoding. Similarly, for each drug SMILES sequence, $D=b_{1},b_{2},\cdots ,b_{n}$, where $b_{i}$ represents the *i*th atom or structure indicator and *m* is the sequence length. We represent each $b_{i}$ as a one-hot encoding, and each original drug embedding is composed of 61 one-hot encodings. Finally, the original embedding of drug and protein sequences is expressed as $D_{e}\in R^{C\times L_{D}}$, $T_{e}\in R^{C\times L_{T}}$, where *C* is the embedding channel size, and $L_{D}$ and $L_{T}$ respectively represent the maximum length of drug and target.

After obtaining the original embedding of the drugs and targets, we input the original embedding vector into the 6-layer CNN to shorten the length of the sequence and reduce computational complexity and time while preserving the original correlation and global information in the sequence.

In order to fully extract the adjacent information in the sequence, we use multiple convolution kernels for each convolution layer to learn about this region embedding. For each sequence, convolution kernels are used for convolution calculation, and each convolution kernel is responsible for extracting a specific segment of information in the sequence. The calculation expression is as follows:(5)$$X_{i}^{\prime }=W_{i}X+b_{i}$$
(6)$$X^{\prime }=\left[X_{1}^{\prime }||X_{2}^{\prime }||\cdots ||X_{N}^{\prime }\right],$$
where $X\in R^{C\times L}$ is the original embedded representation of a given sequence, $W_{i}\in R^{3\times L}$ and $b_{i}\in R^{\left(L-2\right)\times 1}$ respectively denote the weight and bias of the *i*th convolution kernel, *L* is the length of the original embedded representation of drug or protein, $X^{\prime }\in R^{\left(L-2\right)\times N}$ is the representation of a sequence after convolution processing with *N* convolution kernels, || is a concatenation operation.

After that, we used two identical convolution layers to fully extract the information between the sequence region embedding and its adjacent embedding. *N* convolution kernels are used for convolution calculation to extract the interaction information between the specific region embedded in the sequence and its left and right neighbors as below:(7)$$X_{i}^{\prime \prime }=\sum_{k=1}^{n}\left(W_{k,i}\sigma (X_{i}^{\prime }),+,b_{k,i}\right)$$
(8)$$X^{\prime \prime }=[X_{1}^{\prime \prime }||X_{2}^{\prime \prime }||\cdots ||X_{N}^{\prime \prime }],$$
where $X_{i}^{\prime }$ is the *i*th channel of $X^{\prime }$, $W_{k,i}\in R^{3\times L}$ denotes the weight of the *k*th convolution kernel in the *i*th channel of $X^{\prime }$, $b_{k,i}\in R^{\left(L-2\right)\times 1}$ denotes the bias of the *k*th convolution kernel in the *i*th channel of $X^{\prime }$, *σ* is *ReLU* nonlinear activation function, and $X^{\prime \prime }\in R^{\left(L-2\right)\times N}$ is a representation of a sequence after convolution processing with one convolution layer.

Inspired by ResNet [31], we then connect a feature aggregation module consisting of a pooling layer and two convolution layers. We set the pooling layer as the non-linear pooling function of “maximum pooling”, so that the sequence length of sequence feature vectors is reduced by half each time after pooling. After that, the connected convolutional layer is equivalent to linearly weighting the result of the action of nonlinear functions, which strengthens the role of the pooling layer in reducing information redundancy and also reduces the information loss caused by pooling. Finally, we concatenate the pooling results with the convolution results and calculate the expression as follows:(9)$$X^{t+1}=P\left(X^{t}\right)+\theta (P\left(X^{t})\right),$$
when t=0, $X^{t}=X^{\prime \prime \prime }$ and where $X^{\prime \prime \prime }\in R^{\left(L-2\right)\times N}$ denotes the representation of the sequence after the convolution processing of two convolution layers, *P* is the max pooling function, *θ* is the convolution calculation of two convolution layers.

### 3.4. Construct a Heterogeneous Network

In addition to the six original associations in the network, we add additional drug–drug similarity and protein–protein similarity information to the heterogeneous network by considering both biological sequence information and network information.

The network similarity of drug–drug and protein–protein are calculated using the Jaccard similarity coefficient. Specifically, the drug–disease association network, drug–drug interaction network, and drug–side effect association network are used to calculate the similarity measure of each two drug nodes, $D_{1}$ and $D_{2}$, in the network. The formula is shown as follows.
(10)$$J=\frac{N_{11}}{N_{01}+N_{10}+N_{11}},$$
where $N_{11}$ is the total number of nodes that have edges connected to $D_{1}$ and connected to $D_{2}$, $N_{01}$ is the total number of nodes that have edges connected to $D_{1}$ and not connected to $D_{2}$, and $N_{10}$ is the total number of nodes that have edges connected to $D_{2}$ and not connected to $D_{1}$.

In the same way, protein similarity is calculated using the target–disease association network and the target–target interaction network, respectively. The sequence similarity between drugs is calculated based on molecular fingerprints [32,33]. First, MACC fingerprints of molecules in drug sequences are calculated, and the Tanimoto coefficient is obtained based on the similarity comparison of MACC fingerprints. Finally, the Tanimoto coefficient is used to measure the similarity between drugs. MACC fingerprints refer to fingerprints derived from the chemical structure database developed by MDL. A total of 166 substructures are examined, plus 1 bit to hold the information in the RDKit, for a total of 167 bits for the fingerprint. If it has a substructure, store 1, otherwise store 0. The Tanimoto coefficient is an extension of the Jaccard coefficient. The formula is shown as follows.
(11)$$T=\frac{N_{11}}{N_{01}+N_{10}-N_{11}},$$
where $N_{11}$ denotes the number of shared fingerprints of two drugs, $N_{01}$ denotes the total number of fingerprints of $D_{1}$, and $N_{10}$ denotes the total number of fingerprints of $D_{2}$.

The sequence similarity between proteins is calculated using the Levenshtein similarity coefficient. Specifically, we characterize similarity by looking at differences in the length and types of amino acids in different protein sequences as follows:(12)$$I=\left|S_{1}-S_{2}\right|+2E$$
(13)$$L=\frac{S_{1}+S_{2}-I}{S_{1}+S_{2}},$$
where $S_{1}$ is the number of amino acids of $P_{1}$, $S_{2}$ is the number of amino acids of $P_{2}$, and *E* is the sum of the difference between the number of amino acids in $P_{1}$ and $P_{2}$.

For each pair of drugs, we calculate four similarity scores based on network similarity and biological sequence similarity. Therefore, for any pair of drugs, we can obtain four similarity scores. Next, we set a given threshold, and if one of the similarity scores is greater than this threshold and this has no interaction with the drugs, we add a similarity edge between the two drugs. Similarly, for each pair of targets, we calculate three similarity scores based on network similarity and biological sequence similarity. We then determine whether similar edges are added between two targets by a given threshold that we set.

### 3.5. Graph Representation Learning of Drug and Target

One of the key aims of the DTI recognition model is to find a more advanced, better performance, and more reasonable feature extractor. Recently, GCN and GAT have been widely used in feature extraction of graph networks [34,35,36,37]. In order to realize feature aggregation among related nodes in heterogeneous networks, graph convolution is a common method on the network.

In this section, we propose a graph learning module composed of GCN and GAT layers to learn feature representations from heterogeneous networks. As shown in the graph representation learning module in Figure 1, we add a graph attention layer between the two graph convolution layers to help the GCN layer extract high-level features of the drug and target. The following sections explain the details.

We represent the heterogeneous network as $G=\left(V,E,R\right)$, where $V_{i}\in V$ denotes a node in the heterogeneous network (i.e., drugs, proteins). $\left(V_{i}\;,r,V_{j}\right)\in E$ is an edge in the heterogeneous network, and r∈R represents an edge type in the heterogeneous network. Specifically, R includes five types of edges: drug–drug interaction, drug–drug similarity, drug–protein interaction, protein–protein interaction, and protein–protein similarity. In the GCN layer, we aggregate the features among relevant nodes as follows:(14)$$X^{\left(l+1\right)}=\sum_{r}\sigma \left(A_{r}{}^{-\frac{1}{2}}\tilde{S_{r}}A_{r}{}^{-\frac{1}{2}}X^{\left(l\right)}W_{r}^{\left(l\right)}\right),$$
where $S_{r}$ denotes the network adjacency matrix with edge type *r*, $\tilde{S_{r}}=I+S_{r}$. $A_{r}$ is the network degree matrix with edge type *r*, *W* is the trainable weight parameter matrix, $X^{\left(l\right)}$ is the features representation of the node in *l* layer. *σ* represents the *ReLU* activation function. When l=0, we use drug and target biometric feature vectors as original features to encode each node.

In the GAT layer, we aggregate the features among relevant nodes as follows:(15)$$X_{i}^{\left(l+1\right)}=\sum_{r}\sigma \left(\frac{1}{K}\sum_{k=1}^{K}\sum_{j\neq i}\varphi_{r}{}_{ij}^{k}W_{r}{}_{k}^{\left(l\right)}X_{j}^{\left(l\right)}\right)$$
(16)$$\varphi_{r}{}_{ij}^{k}=\frac{\exp\left(L\left(a_{k}^{T}\left[W_{k}X_{i}||W_{k}X_{j}||B_{k}r_{ij}\right]\right)\right)}{\sum_{t\neq i}\exp\left(L\left(a_{k}^{T}\left[W_{k}X_{i}||W_{k}X_{t}||B_{k}r_{it}\right]\right)\right)},$$
where *K* is the number of attention mechanisms in multi-head attention, $\varphi_{r}{}_{ij}^{k}$ denotes the kth attention coefficients between nodes *i* and *j* with edge type *r.*
$W_{k}$ is the weight matrix of the *k*th attention mechanism, $X_{j}^{\left(l\right)}$ is the features representation of node $V_{j}$ in the *l* layer. || is a concatenation operation, *L* is the *LeaklyReLU* activation function. $a_{k}$ denotes the weight vector of the kth attention mechanism, and $B_{k}$ denotes the weight of the edge $r_{ij}$ that is to be learned.

So far, we obtain the feature coding of each node in the heterogeneous network under each layer. For the nodes in the network, the previous models can only aggregate the information of neighbors within one hop by using a layer graph convolution neural network layer number. In order to fully extract the interaction information between nodes and their neighbors in heterogeneous networks, we enlarge the receptive field of the graph by stacking the graph convolution layer and the graph attention layer from the inter-layer perspective, and effectively aggregate the information of multi-hop neighbors [38].

However, stacking multi-layer GCN and GAT may lead to the common problem of feature over-smoothing and the vanishing gradient problem [39,40], that is, the output of the hidden layer representation of each node tends to converge to the same value. In addition, it is inevitable to lose feature information in the feature transmission process between layers. In order to solve the feature over-smoothing and vanishing gradient problems, scholars have put forward some corresponding methods. For example, HGANDTI [41] avoids the problem of over-smoothing features generated by stacking multiple layers by enlarging receptive fields from an intra-layer perspective. LightGCN [42] prevents the problem of missing feature information by considering the output representation of different GCN layers.

Inspired by the above model, we add a fusion layer and combine the output feature vectors of different layers to obtain the final drug and target feature representation $X_{D}\in R^{d_{v}\times d}$ and $X_{T}\in R^{T_{v}\times d}$. The feature aggregation process between different layers of relevant nodes is as follows:(17)$$X^{\left(1\right)}=\mathrm{GAT}\left(S,X^{\left(0\right)}\right)$$
(18)$$X^{\left(2\right)}=\mathrm{GAT}\left(S,X^{\left(1\right)}\right)$$
(19)$$X^{\left(3\right)}=\mathrm{GCN}\left(S,X^{\left(2\right)}\right)$$
(20)$$X=\mathrm{CNN}\left(X^{\left(1\right)},X^{\left(3\right)}\right).$$

In order to train the joint representation learning framework of biological sequences and heterogeneous network features, we use binary cross entropy to measure the gap between the DTI matrix and the preference matrix and use it as a loss function for training the joint representation learning framework. The loss function is shown as follows:(21)$$Loss=-\frac{1}{N}\sum_{\left(i,j\right)}y_{ij}\log\left(u_{ij}\right)+\left(1-y_{ij}\right)\log\left(1-u_{ij}\right),$$
where $y_{ij}$ is the label value between drug *i* and target *j*, $u_{ij}$ is the predicted value between drug *i* and target *j* in the preference matrix $U=X_{D}X_{T}^{T}$, and N=i×j is the number of drug–target pairs.

### 3.6. DTI Prediction

We use the feature joint representation learning framework to connect the drug and target feature representations $X_{D}$ and $X_{T}$ to obtain the fused descriptors of the drug–target pairs. The fusion descriptor for a pair of drug *i* and target *j* is as follows:(22)$$Z_{ij}=\left(X_{D}^{i},X_{T}^{j}\right),$$
where $X_{D}^{i}$ and $X_{T}^{j}$ are the feature representations of drug *i* and target *j*, respectively.

Random forest is an efficient integrated classification algorithm [43]. It generates a list of base evaluators through *N* times of training, and then evaluates its predicted results through the average or voting principle. Thus, overfitting of training data can be effectively alleviated [44]. Currently, it has been widely used to solve problems in bioinformatics fields, such as predicting disease-associated circRNAs [45,46]. Inspired by Lertampaiporn et al. [45], we use fusion descriptors of drug–target pairs ($Z_{ij}$) as categorical features and use a random forest classifier to predict whether DTI has interaction.

## 4. Discussion and Conclusions

In recent years, the prediction of DTI has been of great significance for the discovery of new drugs and the repositioning of drugs. Although the interactions between the drug and target can be accurately verified by biological experiments, it is often a time-consuming and expensive process. Therefore, it is urgent to develop computational models to predict DTI. In this work, we propose a framework based on biological feature and heterogeneous network representation learning for the prediction of DTI, called BG-DTI. Our model combines two mainstream methods, biometric-based and network-based, to extract the characteristics of drugs and targets and to predict the interactions between drugs and targets using the random forest algorithm. We create a benchmark dataset based on the Luo et al. dataset [13], the Yamanishi et al. dataset [28], and the Davis dataset [29], and compare BG-DTI with previous methods on this benchmark dataset. Comparative experiments show that BG-DTI performs better than existing state-of-the-art methods. The ideal predictive ability of BG-DTI mainly depends on the following factors: First of all, none of the previously proposed methods combined biometric features with heterogeneous network methods, while BG-DTI made full use of the sequence feature information of drugs and targets and topological information on the graph through the biological feature extraction module and the graph representation learning module. Second, the predictive process of the previous methods is often fused with the feature extraction module to form the end-to-end DTI prediction model, but these methods also reduce the flexibility and interpretability of the model and sometimes introduce the problem of overfitting. BG-DTI uses random forest as a predictive classifier to obtain a higher quality classification strategy. The results of the ablation experiment show that the drug and target biometric feature extraction module and the graph representation of the learning module can provide more abundant and accurate drug and target information for the prediction of DTI. The random forest algorithm can make the optimal classification decision based on the extracted drug target features, thus improving the prediction performance of DTI [47]. In addition, although BG-DTI is mainly used to predict DTI, it is a portable method and it can be widely used to solve problems in bioinformatics fields such as predicting the correlation between circRNAs and diseases [48,49,50,51].

However, BG-DTI also suffers from some limitations. First, as shown in Figure 2, the performance of BG-DTI under highly unbalanced datasets is not ideal. Second, as shown in Table 4, the performance of BG-DTI on the blind protein test and the benchmark dataset is quite different. The reason for the performance gap may be that the method based on protein sequence extraction means that the model learns insufficient protein structure information. Rational use of protein structure information is an important idea to further improve DTI prediction performance. Third, the prediction of BG-DTI is only aimed at whether the drug target is associated, and there is no detailed prediction of the associations type (e.g., agonist, inhibitor, potentiator, and antagonist) [15]. In the future, we will design and develop a new version of BG-DTI that can perform a more detailed classification of DTI types.

## Figures and Tables

**Figure 1 molecules-28-06546-f001:**
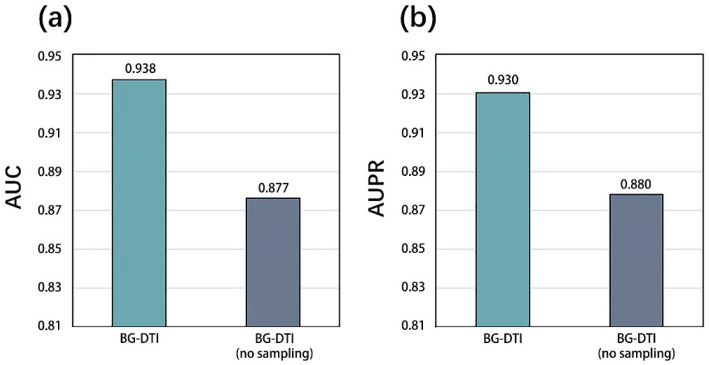
(**a**) AUC results of over-sampling model and no over-sampling model. (**b**) AUPR results of over-sampling model and no over-sampling model.

**Figure 2 molecules-28-06546-f002:**
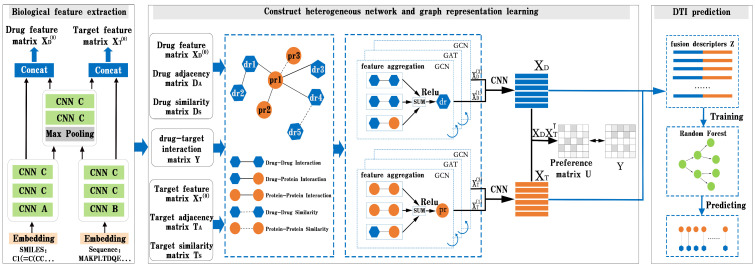
Overview of BG-DTI. The workflow of BG-DTI includes three modules for model construction: drug and target biometric feature extraction module, graph representation of learning module, and DTI prediction module.

**Table 1 molecules-28-06546-t001:** The 5-fold CV testing results of BG-DTI on benchmark dataset.

Validation Set	1	2	3	4	5	Avg.
AUC	0.932	0.945	0.931	0.940	0.943	0.938
AUPR	0.926	0.940	0.929	0.927	0.928	0.930

**Table 2 molecules-28-06546-t002:** Performance comparison of BG-DTI and other methods on the same benchmark dataset.

Model	MultiDTI	NeoDTI	DTINet	HNM	HyperAttentionDTI	BG-DTI
AUC	0.791	0.927	0.914	0.890	0.889	0.938
AUPR	0.855	0.874	0.923	0.572	0.897	0.930

**Table 3 molecules-28-06546-t003:** Comparison of BG-DTI with other methods in terms of three settings from the same benchmark dataset with 5-fold CV.

Model	Blind Drug		Blind Protein		Blind Pair	
	AUC	AUPR	AUC	AUPR	AUC	AUPR
MultiDTI	0.730	0.844	0.709	0.775	0.717	0.801
NeoDTI	0.805	0.753	0.803	0.710	0.797	0.732
DTINet	0.798	0.830	0.795	0.768	0.804	0.818
HNM	0.713	0.576	0.687	0.553	0.737	0.572
HyperAttentionDTI	0.815	0.775	0.811	0.734	0.820	0.753
BG-DTI	0.911	0.887	0.828	0.772	0.904	0.875

**Table 4 molecules-28-06546-t004:** The AUC and AUPR performance on ablation experiments.

Model	AUC	AUPR
BG-DTI	0.938	0.930
DTI-bio	0.881	0.886
DTI-het	0.875	0.889
DTI-sim	0.877	0.880
DTI-rf	0.841	0.723

**Table 5 molecules-28-06546-t005:** Summary of the benchmark dataset.

Types	Items	Numbers
Node	Drug	598
	Target	1352
	Disease	4862
	Side effect	3640
Edge	Drug–drug	7498
	Drug–target	1643
	Disease–drug	173,205
	Drug–side effect	69,300
	Target–target	6206
	Disease–target	1,444,324

## Data Availability

The datasets used in this study and the source code for BG-DTI are publicly available at https://github.com/wyx2012/BG-DTI (accessed on 10 June 2023).

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
