# Peer review of "A Biological Feature and Heterogeneous Network Representation Learning-Based Framework for Drug–Target Interaction Prediction"

_molecules, 2023, doi:10.3390/molecules28186546_

Round 1

Reviewer 1 Report

The review of the manuscript “A biological feature and heterogeneous network representation learning-based framework for drug–target interaction prediction” by Liwei Liu * , Qi Zhang , Yuxiao Wei , Qi Zhao , Bo Liaosubmitted to the the MDPI:Molecules journal

The manuscript “A biological feature and heterogeneous network representation learning-based framework for drug–target interaction prediction” describes a machine learning  (ML) protocol aimed at recognizing protein-ligand (drug-target) interactions. The authors develop ML uses graph convolutional network and graph attention network to extract feature representations of drugs and targets. The manuscript is well written and can easily be grasped (I congratulate the authors on the quality of presentation) I still cannot suggest it for publication. I have a few issues I would like to see addressed:

1.    The authors retrieve interaction data from the interaction data bases, and aim to reconstruct that information, but they never express interaction through standard metrics used in drug design such as binding affinity (kcal/mol) and/or ligand concentration (mm or nm). Binding affinity (binding free energy) is used to express the affinity of a given ligand to the particular target and is invaluable information for drug designers. Without that information the interaction data is insufficient for lead optimization. For example, a ligand can have a binding affinity of -5 kcal/mol, and another ligand can have a much stronger affinity of -11 kcal/mol. Those are totally different affinities and offer different prospects for optimization. Without that information the manuscript seems to be more oriented toward machine learning practitioners, and less to medicinal chemists, or drug designers. If the authors cannot reconstruct that data, they should explain how their tool can be used in real-world drug design setting.

2.    The authors should give more details on their code and ML framework used. I only gathered that they use PyTorch by browsing their GitHub repository. They should reveal that information in the manuscript.

3.    The authors use nonstandard fonts in their equations. They should use more standard fonts, generally used for math equations, and use bold and/or italics to emphasize vectors (matrices).

I believe that if improved, this work can be a very important contribution to the study of drug-target interactions. The authors have tools and enough initial knowledge to improve their research. With additional work and time, this can become a publishable study.

Reviewer 2 Report

The manuscript entitled: "A biological feature and heterogeneous network representation learning-based framework for drug–target interaction prediction" comprises the necessary elements of scientific novelty. This manuscript is well-written. The results are interesting and clearly presented and supported by the literature properly including criticism in discussion and conclusions sections. I recommend that the manuscript should be published.

Reviewer 3 Report

In this manuscript, the authors introduced a machine-learning model to predict whether a drug molecule interacts with a target protein or not. The model takes the SMILES form of the drug molecule, and the protein sequence as input. The first module of the model is comprised of CNN layers. Features of the drug molecule and protein are extracted. In the second module, drug and target proteins are represented as nodes, and interactions and similarity are represented as edges. GCN and GAT layers are stacked for learning. In the final prediction module, the random forest method is used to get an ensemble result. By comparing with other methods on the benchmark dataset, the authors showed that their model has higher AUC and AUPR, and thus is a better model.

I have the following questions/comments:

1.       Can this model be used to predict dissociation constant? The last binary classifier can be changed to predict a number, right? Because drug-protein interaction is not a yes/no question, people care about the affinity.

2.       I agree with the authors that prediction from protein sequence to function is difficult without structural information. One way to overcome this is to use multiple sequence alignment as in AlphaFold2. Have the authors considered that possibility?

3.       It’ll be great if the authors can show a figure of ROC rather than just reporting the AUC.

Reviewer 4 Report

The paper present a new method to calculate the drug-target interactions (DTI) necessary to discover new drug. The article is well structured and applied new methologies to obtain good results. 

The only change I propose is about the Figure 2. It is very small and is very difficult to appreciate all the information presented. 
